# Plant Protection and Fertilizer Use Efficiency in Farms in a Context of Overinvestment: A Case Study from Poland

Jagoda Zmyślona *[ID], Arkadiusz Sadowski and Natalia Genstwa

Faculty of Economics, Department of Economics and Economic Policy in Agribusiness, Poznan University of Life Sciences, 60-637 Poznan, Poland; arkadiusz.sadowski@up.poznan.pl (A.S.); natalia.genstwa@up.poznan.pl (N.G.)
* Correspondence: jagoda.zmyslona@up.poznan.pl

**Abstract:** The purpose of this study was to determine the relationship between plant protection and fertilizer use efficiency, on one side, and overinvestment in Polish agriculture, on the other. This is an important topic because of a number of essential issues, such as the concern for the environment, the development of sustainable agriculture, or the need to ensure food security which can only be achieved by keeping production volumes at least at the same level. Reconciling these goals often requires investment which, however, involves the risk of overinvesting, i.e., a situation where the value of assets grows without a proportional increment in labor productivity. This paper uses the author's own method of farm classification by overinvestment level. The study revealed some differences in the cost intensity of fertilizing and using plant-protection products between investment levels. The most rational results were found in farms at optimum investment levels, whereas the greatest cost intensity was recorded in farms affected by overinvestment.

**Keywords:** overinvestment; agriculture; environment; plant protection; fertilizer

## 1. Introduction

Meeting the need for food remains the basic human need [1–3]. The growing population made it necessary to intensify agriculture, which largely means an increased use of fertilizers and plant-protection products [4–7]. According to the Fertilizers Europe report [8], as much as 50% of the world's population is provided with food produced with the use of agricultural fertilizers. The above makes fertilizers an essential part of ensuring food security, and a crucial yield booster [9]. A specific feedback loop exists in that context—the use of industrial yield boosters contributes to population growth which, in turn, makes it necessary to increase production, mostly by using these very industrial products [10]. The whole process has an impact on the climate and the environment [11–13].

The climate is threatened by greenhouse-gas emissions, one of which is nitrous oxide [14], whose calorific potential is more than 300 times that of carbon dioxide [15]. It is released into the atmosphere either directly or through denitrification, a process resulting from improper agricultural practices and excessive fertilization rates [16,17]. Equally important is the aspect related to surface and groundwater pollution, which has an adverse effect on biodiversity [18]. Also, it impacts the chemical composition of potable water which, in view of its shrinking resources, represents a major problem [19,20]. Moreover, improper fertilization may reduce soil fertility, i.e., become counterproductive [20,21]. That problem is noticeable in Europe, where plow tillage plays a dominant role, and agricultural land accounts for nearly half of most countries' landmass. Hence, a conflict arises between the irreducible need to ensure food security for the growing population and the equally important environmental and climate goals [22,23]. As the main carbon sink, soils play an important role in reducing greenhouse-gas emissions [24] by providing and regulating key ecosystem services and ensuring biodiversity protection [25]. However, in Europe, the soil-loss rate is almost twice the soil-formation rate [26,27]. It should be noted that,

of course, agriculture is not the only sector contributing to climate change. Nevertheless, its contribution to greenhouse-gas emissions is around 22% (together with forestry and other land use) [28]. In the context of the use of fertilizers and plant protection, it is also important to mention their impact on the local environment, mainly on soil quality [11]. It can be both positive and negative, depending on the rationality of the application [4]. The implementation of good practices can even improve soil quality relative to its natural state [29]. The subject is particularly important in the case of Polish agriculture, which is characterized, compared to the EU countries, by a relatively high cost of fertilization and plant protection in relation to the volume of production. This is because the use of a chemical yield booster is a kind of substitute for failure in other agrotechnical practices (such as the use of certified seed) [9].

That issue could be solved by rationalized fertilizer use, so as to address the need for food while minimizing the adverse environmental impacts [30]. A similar problem arises with respect to plant-protection products. Finally, the use of yield boosters also has an economic dimension. Indeed, the farms struggle with what is referred to as "margin squeeze": the price relationship between agricultural raw materials and productive inputs such as seed, seedlings, energy, fertilizers, plant-protection products, or feedingstuffs [31]. During extreme price surges, the farmers must sometimes decide to discontinue using some productive inputs. But on the other hand, the growing population means that agricultural production must grow.

The above has become even more important due to the Russian attack on Ukraine [32]. On the one hand, this makes the ever-existing problem of food security even more alarming [32,33], but, on the other, the rising prices make it even more imperative to seek rationalization of pesticide and fertilizer use. As regards the latter, note the convergence between environmental requirements and microeconomic goals of agricultural producers who seek maximization of economic results.

In an attempt to solve the conflicts between the need to produce food and the need to protect the environment [11], European Union countries came up with the European Green Deal (EGD) [34] concept, which is supposed to make the EU a modern, resource-efficient, competitive economy, so as to reduce net greenhouse-gas emissions to zero by 2050 [24,35]. It spans all sectors, including agriculture, which is particularly addressed in the "Farm to Fork" and "Biodiversity" strategies [36]. In the latter, soil plays a key role in meeting the goals of sustainable development [37]. The EGD assumes a 50% reduction in the use of chemical plant-protection products (and related risks), a 20% or greater reduction in the use of artificial fertilizers, and a 50% reduction in the use of antibiotics by 2030 [38]. Also, organic production is expected to grow and is supposed to account for 25% of agricultural land. The EGD sets objectives for biodiversity, climate change, sustainable agriculture, and rural development [39]. It emphasizes the importance of reducing soil pollution [40,41] caused by large quantities of fertilizers being used in agriculture [42]. As regards agriculture, the essential shortcoming of the EGD is the absence of clearly defined methods for seeking these goals and its failure to address the need for ensuring food security. The problem is mostly about reducing the use of industrial yield boosters; counteracting climate change cannot become a growth barrier for food production [43]. According to a number of studies [9,44,45], the reduction in the use of nitrogenous fertilizers is particularly likely to contribute to a reduction in agricultural production volumes. Hence, EGD assumptions make it necessary once again to rationalize the use of industrial yield boosters. In order to reduce their use with no adverse effect on yields, it is essential to employ adequate agrotechnical measures, on the one hand, and to invest.

Investments can be economically unviable, which means the economic operator suffers a loss in the long term. This process can also be noticed in the agricultural sector and affects specific groups of farms to various degrees. This results in the emergence of what is referred to as overinvestment, i.e., a condition where long-term investments are excessively high compared to the production potential (mainly land resources) and, ultimately, become economically unviable [46]. Optimum investment means a situation where the assets-to-

labor ratio (*ALR*) of a farm grows at the same pace as labor productivity (*LP*) [47]. In turn, overinvestment takes place when the assets-to-labor ratio grows while labor productivity declines or remains constant [46]. The inefficiency of investments brings many negative effects to farms. In addition to achieving worse economic results, we can observe behaviors in the very decision-making processes regarding the optimization of the production path, including the efficiency of the use of fertilizers and plant protection products.

Investigating the relationship between overinvestment and the use efficiency of fertilizers and plant-production products is all the more important since—as mentioned earlier in this paper—investments are necessary (in addition to agrotechnical measures and agricultural knowledge) in order to rationalize fertilization. As demonstrated by Wei et al. [48], the larger the farm, the better the fertilizer use efficiency. Yang and Lin [49] pointed to the rationale of subsidizing investments in the agricultural sector as more efficient for farms and the environment than subsidies for fertilizer application. In addition, climate variability means that investments in fertilizers and crop-protection products alone may not yield the expected results [50]. This results from optimum fertilization levels which, in practice, means minimizing the fertilizer-to-yields ratio (while reducing environmental pollution). Indeed, the owners of larger farms are more aware of how to use fertilizers and rely on more sophisticated technical equipment [51]. Therefore, Stoicea et al. [52] suggest that reduction rates should be set on a per-country basis because of differences in development levels. Nevertheless, the investments themselves are important in achieving optimal levels of fertilizer and crop protection-product use. At the same time, however, there is the problem of hasty investments leading to overinvestment, which interferes with optimal decision-making, including in the area of fertilizer and pesticide application.

As a consequence, having in mind the future challenges and problems involved in the implementation of the assumptions behind the European Green Deal, as described above, the purpose of this study was to determine the relationship between plant protection and fertilizer use efficiency, on one side, and overinvestment in Polish agriculture, on the other.

As agriculture is a high-emission sector, the environmental efficiency of food production should be a major element of the policy designed to counteract climate change [53]. Thus, investment subsidies should also contribute to the general social goal of protecting the environment and climate [54].

## 2. Materials and Methods

The research materials were unpublished 2010–2019 accounting data for 3273 Polish farms, retrieved from the FADN. (FADN is a farming accountancy system used in all EU member countries. It covers commercial farms which jointly account for 90% of a country's standard output and is mostly used to monitor the effects of the Common Agricultural Policy and to develop its future assumptions) database. Farms were divided into investing farms (investing with investment support and investing without investment support) and noninvesting farms. For this purpose, those farms for which the sum of investments in 2010–2019 was less than PLN 10,000 were considered noninvesting. According to accounting records on fixed assets, depreciation is made on assets above the value of PLN 10,000. On the other hand, farms using investment support were considered those for which the sum of investment support for the surveyed 10 years was greater than PLN 0. All others were considered nonusers of investment support. The research diagram is shown in Figure 1.

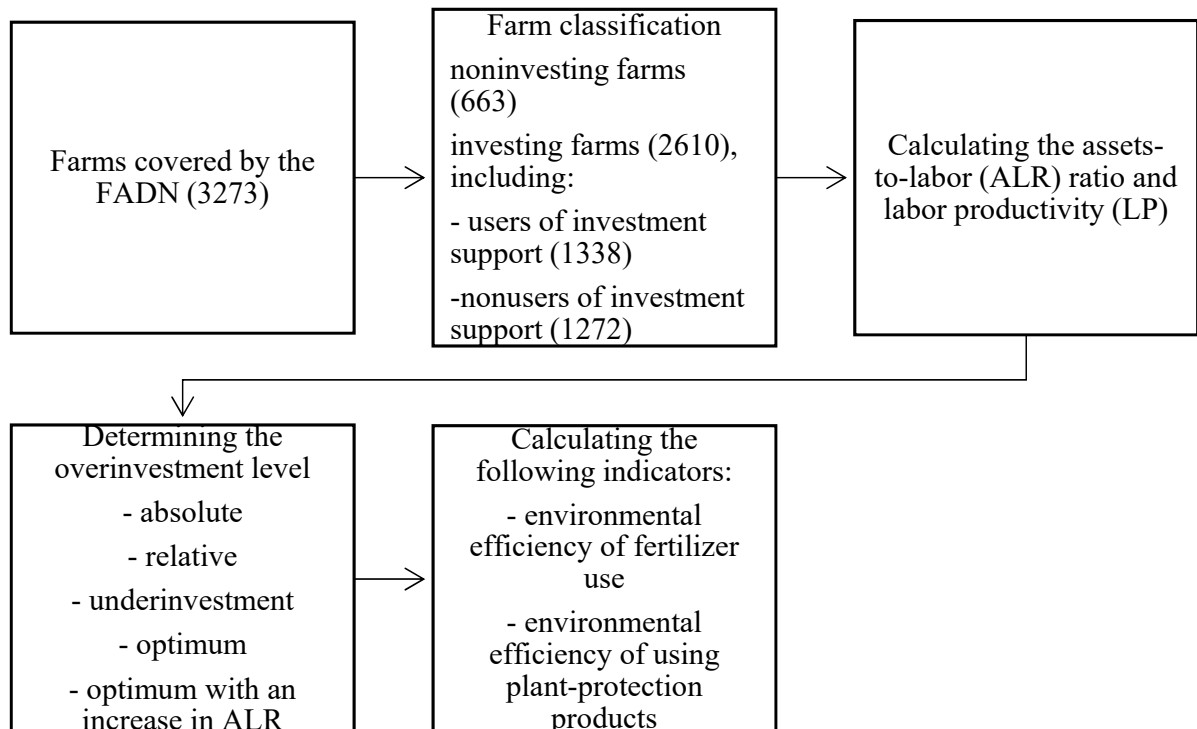

**Figure 1.** Conceptual Framework (the number of farms covered by the analysis is specified in brackets). Source: own compilation.

In order to enable a more precise comparison, the farms were divided into noninvesting and investing ones, which, in turn, were grouped as users and nonusers of EU investment support (Figure 1).

This paper assumes that increasing the value of farm assets through investments is a reasonable thing to do if it results in a proportional growth in labor productivity. Therefore, overinvestment is defined as a situation where:

- The increase in the value of assets results in a decline or stagnation in labor productivity, which may be due to the high maintenance costs of particular assets (e.g., depreciation, insurance, and repairs). The above is defined as absolute overinvestment;
- Labor productivity grows at a lower rate than the value of assets. This is referred to as relative overinvestment.

The study period spanned from 2010 to 2019; the growth rates for asset amounts and labor productivity (of essential importance for the goal of this study) were determined for the final year.

The noninvesting farms were only split into underinvested and other holdings because farms that did not make any investments over a 10-year period cannot find themselves at excessive or optimum investment levels. Average values could be calculated by creating panel data based on microdata. Thus, it was possible to initially identify a single type of overinvestment for the whole group of farms covered by the study, based on whether or not they accessed investment subsidies. Also, overinvestment levels were calculated separately for each farm. It turned out that the database included farms that shifted between overinvestment levels over the study years. Therefore, it was decided that the best way to determine their overinvestment status would be to identify five separate 2-year periods and to compare the last period against the baseline. This resulted in attributing a specific overinvestment level to each farm based on the last period covered by the study (2018–2019). Also, using two-year average figures allowed for the avoidance of price fluctuations.

The study used a previously employed method [55]. Labor productivity was calculated as gross value added less depreciation, less investment subsidy installments, and less operating subsidies per employee:

$$LP_t = \frac{\sum_t^{t+1} \left( \frac{SE410 - SE360 - SE406 - SE605}{SE010} \right)}{2} \tag{1}$$

$$\Delta LP = \left( \frac{LP_{t5} - LP_{t0}}{LP_{t0}} \right) * 100\% \tag{2}$$

where:
   *LP*: labor productivity;
   *SE*410: gross value added;
   *SE*360: depreciation;
   *SE*406: investment subsidy installments;
   *SE*605: operating subsidies;
   *SE*010: total labor inputs (AWU).
   As the next step, the assets-to-labor ratio was calculated as per the following formula:

$$ALR_t = \frac{\sum_t^{t+1} \left( \frac{SE441 - SE446}{SE010} \right)}{2} \tag{3}$$

$$\Delta ALR = \left( \frac{ALR_{t5} - ALR_{t0}}{ALR_{t0}} \right) * 100\% \tag{4}$$

where:
   *ALR*: assets-to-labor ratio;
   *SE*441: fixed assets;
   *SE*446: land, permanent crops, and production quotas;
   *SE*010: total labor inputs (AWU).
   After calculating the two parameters necessary to determine investment levels, the farm data was distributed between the groups in accordance with the author's own methodology:
   I. Absolute overinvestment: this is the case for farms where labor productivity drops while the assets-to-labor ratio grows:

$$\Delta LP < 0 \qquad \wedge \qquad \Delta ALR > 0$$

II. Relative overinvestment: this is the case for farms where both labor productivity and the assets-to-labor ratio are on an increase but the increase in the assets-to-labor ratio is greater than the increase in labor productivity:

$$\Delta LP > 0 \qquad \wedge \qquad \Delta ALR > 0 \qquad \wedge \qquad \Delta LP < \Delta ALR$$

III. Underinvestment: this is the case for farms where both labor productivity and the assets-to-labor ratio are on a decline:

$$\Delta LP < 0 \qquad \wedge \qquad \Delta ALR < 0$$

IV. Optimum investments: this is the case for farms where both labor productivity and the assets-to-labor ratio are on an increase, and labor productivity grows faster than the assets-to-labor ratio:

$$\Delta LP > 0 \qquad \wedge \qquad \Delta ALR > 0 \qquad \wedge \qquad \Delta LP > \Delta ALR$$

V. Optimum investments with no increase in the assets-to-labor ratio: this is the case for farms where labor productivity grows while the assets-to-labor ratio neither grows nor declines:

$$\Delta LP > 0 \qquad \wedge \qquad \Delta ALR < 0 \qquad \wedge \qquad \Delta LP > \Delta ALR$$

The next step consisted of calculating the average consumption level of fertilizers and plant-protection products for each overinvestment level in each investment group. This allowed for the determination of the environmental efficiency of using fertilizers and plant-protection products in the farms covered by the study. The quantity and efficiency of fertilizer use was calculated as per the following formulas:

$$FC = \frac{\sum F}{\sum L} \tag{5}$$

$$FE = \frac{\sum F}{\sum P} \tag{6}$$

where:

*FC*: fertilization costs;
*FE*: fertilization efficiency;
*F*: fertilization costs (PLN);
*P*: production value (PLN);
*L*: farmland area (ha of agricultural land).

The amount and efficiency of plant-protection products used were calculated in a similar way:

$$PPC = \frac{\sum SCPP}{\sum L} \tag{7}$$

$$EEPP = \frac{\sum SCPP}{\sum P} \tag{8}$$

where:

*PPC*: plant-protection costs;
*EEPP*: environmental efficiency of plant-protection products used;
*SCPP*: consumption scale of plant-protection products (PLN);
*P*: output (PLN);
*L*: farmland area (ha).

The analysis of results made it possible to indicate the optimum development path for farm investments while taking their environmental impact into account.

## 3. Results and Discussion

In order to determine the changes in the use of fertilizers and plant-protection products, it is necessary to identify the changes in average land resources (Table 1). This is because the use and efficiency of industrial productive inputs depend on a number of factors, including the farms' technical equipment, which, in turn, varies in function of farm size.

Land resources owned by a farm are largely decisive for its competitive position and capacity to generate incomes. Changes in land resources can be found in farms grouped by investment amount and by overinvestment level. In each group, underinvested farms and those which make optimum investments with no increase in the assets-to-labor ratio were found to have the smallest area. Conversely, the physically largest farms form part of the absolute and relative overinvestment groups. In general, the noninvesting group saw a decline in their average farm area. However, land resources of the "other" group decreased at a faster rate. These trends could be indicative of the smallest farmers believing that they have no competitive advantages. Such a relationship is probably due to the farms gradually discontinuing their agricultural production activities and making more and more divestments [56].

**Table 1.** Average farm area (ha per farm) grouped by overinvestment level and propensity to invest in 2010–2019.

| Period | T0 | T1 | T2 | T3 | T4 | T4 − T0 | T0 = 100% |
|---|---|---|---|---|---|---|---|
| noninvesting farms | | | | | | | |
| underinvested | 21.33 | 20.91 | 20.18 | 19.92 | 19.07 | −2.26 | −11% |
| other | 26.98 | 22.37 | 21.70 | 20.91 | 20.10 | −6.88 | −26% |
| farms that invest and access investment support | | | | | | | |
| underinvested | 49.82 | 51.78 | 53.16 | 53.52 | 53.85 | 4.03 | 8% |
| optimum investment, no increase in *ALR* | 35.27 | 37.52 | 39.49 | 42.07 | 42.89 | 7.62 | 22% |
| optimum investment, increase in *LP* and *ALR* | 64.8 | 63.84 | 66.03 | 69.41 | 71.84 | 7.04 | 11% |
| absolute | 55.45 | 57.50 | 59.34 | 61.67 | 63.28 | 7.83 | 14% |
| relative | 67.97 | 69.4 | 68.39 | 68.83 | 70.34 | 2.37 | 3% |
| farms which do not access investment support | | | | | | | |
| underinvested | 33.87 | 33.71 | 33.76 | 33.92 | 33.82 | −0.05 | 0% |
| optimum investment, no increase in *ALR* | 30.35 | 30.59 | 31.51 | 33.03 | 34.25 | 3.90 | 13% |
| optimum investment, increase in *LP* and *ALR* | 28.44 | 30.12 | 31.24 | 33.40 | 35.77 | 7.33 | 26% |
| absolute | 52.31 | 52.85 | 51.30 | 51.22 | 50.81 | −1.50 | −3% |
| relative | 82.67 | 83.92 | 84.24 | 85.44 | 87.42 | 4.75 | 6% |

Source: own compilation based on the FADN microdata base, n = 3273.

In the group of investing farms that access investment support, members of the "absolute overinvestment" subgroup saw their land resources growing at a rate comparable to that of farms at optimum investment levels. In turn, the slowest growth was recorded in farms classified as "relative overinvestment". As regards the investing farms which do not access investment support, the fastest growth in land resources was witnessed in holdings which recorded an increase in the assets-to-labor ratio while being at optimum investment levels. Conversely, a decline in land area was found in members of the "absolute overinvestment" subgroup and, though to a minimum extent, in underinvested farms. A small growth or decline in the land area of investing farms could mean they made excessive investments in fixed assets and therefore did not have the financial capacity to increase their land resources, or that no one offered land for sale. Also, they predominantly followed a capital-intensive development path.

The cost intensity of fertilization was found to grow considerably in underinvested farms but remained constant in other holdings (Table 2). As a consequence, the fertilization efficiency varied quite strongly in the final years of the study, which may indirectly result from the lack of technical equipment and from an improper use of fertilizers. In 2019, underinvested farms incurred a cost of PLN 0.14 per PLN 1 worth of production, compared to as little as PLN 0.10 in other holdings.

Environmental efficiency of plant protection use remained at a constant level in both underinvested and other farms. They were quite small and similar in both groups.

Despite fertilization being less cost intensive in the "other" group, the amount of the costs per area unit (Table 3) was higher in all years. This could be an indirect indication that relatively high (though rational from an economic and environmental perspective) fertilization rates can be closer to an optimum relationship between inputs and outputs. Over subsequent years of the study, both groups experienced a certain increase in fertilization costs per hectare of agricultural land (33% for underinvested and 18% for other farms). Similarly, plant-protection costs per hectare were higher in other farms throughout the study period but followed a downward trend (a decline of more than 20% between 2020 and 2019). The cost intensity of plant protection was smaller in underinvested farms each year but remained at a near-constant level during the whole study period.

**Table 2.** Relationship between plant fertilization and protection costs, and production value (*FE* and *EEPP*) in noninvesting farms in 2010–2019 (PLN).

| | Fertilization | |
|---|---|---|
| **Year** | **Underinvested** | **Other** |
| 2010 | 0.09 | 0.09 |
| 2011 | 0.09 | 0.11 |
| 2012 | 0.10 | 0.09 |
| 2013 | 0.11 | 0.11 |
| 2014 | 0.12 | 0.10 |
| 2015 | 0.14 | 0.12 |
| 2016 | 0.13 | 0.10 |
| 2017 | 0.12 | 0.09 |
| 2018 | 0.13 | 0.10 |
| 2019 | 0.14 | 0.10 |
| 2010 = 100 | 152 | 110 |
| | **Plant Protection** | |
| **Year** | **Underinvested** | **Other** |
| 2010 | 0.04 | 0.05 |
| 2011 | 0.04 | 0.05 |
| 2012 | 0.04 | 0.04 |
| 2013 | 0.04 | 0.04 |
| 2014 | 0.05 | 0.05 |
| 2015 | 0.05 | 0.04 |
| 2016 | 0.05 | 0.04 |
| 2017 | 0.04 | 0.04 |
| 2018 | 0.05 | 0.04 |
| 2019 | 0.05 | 0.04 |
| 2019 = 100 | 118 | 73 |

Source: own compilation based on the FADN microdata base, n = 3273.

As regards users of investment support (Table 4), fertilization efficiency and environmental efficiency of plant protection use were found to have decreased (by 19% and 24%, respectively) in farms at optimum investment levels. Conversely, an increase in these costs was recorded in members of the "absolute overinvestment" group and in underinvested holdings. As shown by previous research [55], farms that demonstrate absolute overinvestment also prove to be the least technically efficient ones. It is particularly interesting that the differences in fertilization efficiency and environmental efficiency of plant protection use between groups grow over the years. In 2019, the highest levels were recorded both in underinvested farms and in members of the "absolute overinvestment" group. Conversely, the most beneficial values are found on farms at optimum investment levels with no increase in the assets-to-labor ratio, and in optimum holdings. It can therefore be seen that (just like in Table 2) rational investment goes hand in hand with rational production. This may be due to several reasons. First, it could be the consequence of the owners having generally greater economic and agricultural knowledge and being able to concurrently invest and run production processes in a skillful way. As regards underinvested farms, that condition can also result from the absence of adequate equipment. The above is not true for members of the "absolute overinvestment" group which, by definition, have access

to equipment. However, they make inefficient use of it, at least when it comes to plant fertilization and protection.

**Table 3.** Average plant fertilization and protection costs in relation to land area owned (*FC* and *PPC*) by noninvesting farms in 2010–2019 (PLN/ha).

| Fertilization | | |
|---|---|---|
| Year | Underinvested | Other |
| 2010 | 433 | 534 |
| 2011 | 503 | 639 |
| 2012 | 576 | 616 |
| 2013 | 597 | 660 |
| 2014 | 616 | 618 |
| 2015 | 626 | 652 |
| 2016 | 568 | 617 |
| 2017 | 566 | 590 |
| 2018 | 573 | 597 |
| 2019 | 578 | 633 |
| 2010 = 100 | 133 | 118 |
| **Plant Protection** | | |
| Year | Underinvested | Other |
| 2010 | 187 | 305 |
| 2011 | 205 | 304 |
| 2012 | 213 | 250 |
| 2013 | 229 | 272 |
| 2014 | 246 | 269 |
| 2015 | 214 | 243 |
| 2016 | 213 | 234 |
| 2017 | 211 | 245 |
| 2018 | 201 | 220 |
| 2019 | 193 | 240 |
| 2010 = 100 | 103 | 79 |

Source: own compilation based on the FADN microdata base, n = 3273.

As regards investment-support users, fertilization costs follow a clear upward trend at each investment level (Table 5). However, there were quite considerable differences between them every year. Each time, the highest values were found in the "relative overinvestment" group, and the lowest in farms at optimum investment levels with no increase in the assets-to-labor ratio. In the latter case, however, low cost-intensity levels go hand in hand with the smallest fertilization value per hectare.

Plant-protection costs grow at all levels considered in this study (with the fastest rates being recorded in the "absolute overinvestment" and "underinvestment" groups) and differ between the levels. Just like in the case of fertilization, the greatest plant-protection costs per hectare are borne by members of the relative overinvestment group and the smallest by holdings at optimum investment levels with no increase in the assets-to-labor ratio.

As regards nonusers of investment support, fertilization efficiency, that is the ratio between fertilization costs and production (Table 6), grows only in members of the "absolute overinvestment" group (by 36%) and in underinvested farms (by 44%). In the "relative overinvestment" class, it dropped by 14% and remained unchanged in other groups. Nev-

ertheless, throughout the study period, the greatest cost intensity was found in members of the absolute and relative overinvestment groups, and the smallest in optimum farms (whether with or without an increase in the assets-to-labor ratio). This is similar to what was witnessed in users of investment support. Once again, there is a noticeable correlation between strategic and ongoing rationality. Holdings that invest either too little or too much demonstrate the highest—and consistently growing—ratio between fertilization costs and production value. In turn, optimum investors know how to optimize that variable (at least compared to what is seen in other farms covered by this study) and keep it at low levels. Moreover, just like in users of investment support, optimum farms (whether with or without an increase in the assets-to-labor ratio) attained high levels of fertilization efficiency at the relatively lowest costs per hectare of agricultural land (Table 7). While a similar pattern is true for the environmental efficiency of plant protection use, that is a relationship between plant-protection costs and production value, and only members of the "absolute overinvestment" group saw a growth rate of 17%. However, in this case, the highest costs are incurred in the absolute and relative overinvestment groups and the smallest (around half smaller) in farms at optimum investment levels. Hence, this is yet another example of a correlation between strategic and operational rationality.

**Table 4.** Relationship between plant fertilization and protection costs and production value (*FE* and *EEPP* in investing farms that accessed investment support in 2010–2019 (PLN).

| Fertilization | | | | | |
|---|---|---|---|---|---|
| Year | Underinvested | Optimum | Optimum Investment with No Increase in *ALR* | Absolute | Relative |
| 2010 | 0.10 | 0.10 | 0.06 | 0.11 | 0.10 |
| 2011 | 0.10 | 0.10 | 0.07 | 0.11 | 0.12 |
| 2012 | 0.11 | 0.10 | 0.07 | 0.12 | 0.11 |
| 2013 | 0.12 | 0.10 | 0.07 | 0.12 | 0.12 |
| 2014 | 0.13 | 0.10 | 0.08 | 0.14 | 0.13 |
| 2015 | 0.13 | 0.11 | 0.08 | 0.15 | 0.13 |
| 2016 | 0.14 | 0.10 | 0.08 | 0.15 | 0.14 |
| 2017 | 0.12 | 0.09 | 0.07 | 0.13 | 0.11 |
| 2018 | 0.13 | 0.08 | 0.07 | 0.14 | 0.10 |
| 2019 | 0.13 | 0.08 | 0.07 | 0.14 | 0.10 |
| 2010 = 100 | 135 | 81 | 114 | 121 | 96 |
| Plant Protection | | | | | |
| Year | Underinvested | Optimum | Optimum Investment with No Increase in *ALR* | Absolute | Relative |
| 2010 | 0.05 | 0.05 | 0.03 | 0.05 | 0.07 |
| 2011 | 0.04 | 0.05 | 0.03 | 0.05 | 0.07 |
| 2012 | 0.04 | 0.04 | 0.03 | 0.05 | 0.06 |
| 2013 | 0.05 | 0.04 | 0.03 | 0.06 | 0.06 |
| 2014 | 0.06 | 0.04 | 0.03 | 0.06 | 0.07 |
| 2015 | 0.06 | 0.04 | 0.03 | 0.06 | 0.08 |
| 2016 | 0.06 | 0.04 | 0.03 | 0.07 | 0.07 |
| 2017 | 0.06 | 0.04 | 0.03 | 0.06 | 0.07 |
| 2018 | 0.06 | 0.03 | 0.03 | 0.07 | 0.05 |
| 2019 | 0.06 | 0.04 | 0.03 | 0.06 | 0.05 |
| 2010 = 100 | 123 | 74 | 95 | 118 | 78 |

Source: own compilation based on the FADN microdata base, n = 3273.

**Table 5.** Average plant fertilization and protection costs in relation to land area owned (*FC* and *PPC*) by investing farms that accessed investment support in 2010–2019 (PLN/ha).

| | | | Fertilization | | |
|---|---|---|---|---|---|
| **Year** | **Underinvested** | **Optimum** | **Optimum Investment with No Increase in *ALR*** | **Absolute** | **Relative** |
| 2010 | 557 | 648 | 496 | 625 | 713 |
| 2011 | 665 | 759 | 586 | 714 | 843 |
| 2012 | 804 | 928 | 657 | 872 | 938 |
| 2013 | 838 | 1005 | 745 | 846 | 1070 |
| 2014 | 859 | 941 | 741 | 919 | 1087 |
| 2015 | 830 | 976 | 739 | 911 | 1078 |
| 2016 | 849 | 912 | 673 | 908 | 1078 |
| 2017 | 802 | 872 | 707 | 838 | 965 |
| 2018 | 783 | 859 | 697 | 823 | 929 |
| 2019 | 832 | 914 | 766 | 860 | 966 |
| 2010 = 100 | 149 | 141 | 154 | 138 | 135 |
| | | | Plant Protection | | |
| **Year** | **Underinvested** | **Optimum** | **Optimum Investment with No Increase in *ALR*** | **Absolute** | **Relative** |
| 2010 | 279 | 322 | 230 | 296 | 469 |
| 2011 | 287 | 362 | 249 | 335 | 490 |
| 2012 | 315 | 399 | 251 | 347 | 509 |
| 2013 | 342 | 393 | 279 | 388 | 547 |
| 2014 | 376 | 419 | 289 | 407 | 600 |
| 2015 | 343 | 381 | 273 | 373 | 609 |
| 2016 | 375 | 382 | 259 | 403 | 558 |
| 2017 | 384 | 389 | 257 | 408 | 567 |
| 2018 | 355 | 351 | 264 | 400 | 472 |
| 2019 | 379 | 412 | 295 | 395 | 514 |
| 2010 = 100 | 136 | 128 | 129 | 133 | 109 |

Source: own compilation based on the FADN microdata base, n = 3273.

**Table 6.** Relationship between plant fertilization and protection costs and production value (*FE* and *EEPP*), in investing farms that did not access investment support in 2010–2019 (PLN).

| | | | Fertilization | | |
|---|---|---|---|---|---|
| **Year** | **Underinvested** | **Optimum** | **Optimum Investment with No Increase in *ALR*** | **Absolute** | **Relative** |
| 2010 | 0.09 | 0.08 | 0.08 | 0.11 | 0.14 |
| 2011 | 0.09 | 0.08 | 0.08 | 0.12 | 0.15 |
| 2012 | 0.09 | 0.08 | 0.09 | 0.13 | 0.18 |
| 2013 | 0.10 | 0.09 | 0.09 | 0.14 | 0.18 |
| 2014 | 0.11 | 0.09 | 0.10 | 0.14 | 0.14 |
| 2015 | 0.11 | 0.09 | 0.10 | 0.16 | 0.14 |
| 2016 | 0.11 | 0.09 | 0.09 | 0.15 | 0.11 |
| 2017 | 0.10 | 0.08 | 0.08 | 0.14 | 0.12 |
| 2018 | 0.11 | 0.08 | 0.08 | 0.15 | 0.10 |
| 2019 | 0.13 | 0.08 | 0.08 | 0.15 | 0.12 |
| 2010 = 100 | 144 | 100 | 100 | 136 | 86 |

**Table 6.** *Cont.*

| | | | Plant Protection | | |
|---|---|---|---|---|---|
| Year | Underinvested | Optimum | Optimum Investment with No Increase in *ALR* | Absolute | Relative |
| 2010 | 0.05 | 0.03 | 0.04 | 0.06 | 0.06 |
| 2011 | 0.04 | 0.03 | 0.04 | 0.06 | 0.06 |
| 2012 | 0.04 | 0.03 | 0.04 | 0.06 | 0.05 |
| 2013 | 0.04 | 0.03 | 0.04 | 0.06 | 0.06 |
| 2014 | 0.05 | 0.03 | 0.05 | 0.07 | 0.05 |
| 2015 | 0.05 | 0.03 | 0.04 | 0.07 | 0.05 |
| 2016 | 0.05 | 0.03 | 0.04 | 0.07 | 0.04 |
| 2017 | 0.04 | 0.03 | 0.04 | 0.07 | 0.05 |
| 2018 | 0.05 | 0.03 | 0.04 | 0.07 | 0.04 |
| 2019 | 0.05 | 0.03 | 0.04 | 0.07 | 0.05 |
| 2010 = 100 | 100 | 100 | 100 | 117 | 83 |

Source: own compilation based on the FADN microdata base, n = 3273.

**Table 7.** Average plant fertilization and protection costs in relation to land area owned (*FC* and *PPC*) by investing farms that did not access investment support in 2010–2019 (PLN/ha).

| | | | Fertilization | | |
|---|---|---|---|---|---|
| Year | Underinvested | Optimum | Optimum Investment with No Increase in *ALR* | Absolute | Relative |
| 2010 | 502 | 522 | 490 | 584 | 692 |
| 2011 | 573 | 586 | 574 | 702 | 933 |
| 2012 | 658 | 695 | 681 | 862 | 1114 |
| 2013 | 711 | 781 | 733 | 876 | 1086 |
| 2014 | 705 | 747 | 743 | 845 | 915 |
| 2015 | 691 | 771 | 691 | 917 | 850 |
| 2016 | 653 | 713 | 662 | 884 | 749 |
| 2017 | 642 | 697 | 660 | 866 | 756 |
| 2018 | 670 | 726 | 624 | 844 | 730 |
| 2019 | 745 | 826 | 689 | 901 | 762 |
| 2010 = 100 | 148 | 158 | 141 | 154 | 110 |
| | | | Plant Protection | | |
| Year | Underinvested | Optimum | Optimum Investment with No Increase in *ALR* | Absolute | Relative |
| 2010 | 251 | 209 | 256 | 293 | 315 |
| 2011 | 259 | 227 | 272 | 335 | 374 |
| 2012 | 286 | 254 | 271 | 383 | 338 |
| 2013 | 305 | 264 | 294 | 398 | 337 |
| 2014 | 305 | 276 | 347 | 423 | 347 |
| 2015 | 295 | 263 | 294 | 411 | 329 |
| 2016 | 287 | 273 | 275 | 394 | 276 |
| 2017 | 286 | 282 | 320 | 405 | 333 |
| 2018 | 292 | 261 | 289 | 365 | 287 |
| 2019 | 291 | 266 | 295 | 381 | 325 |
| 2010 = 100 | 116 | 128 | 115 | 130 | 103 |

Source: own compilation based on the FADN microdata base, n = 3273.

The analysis of nonusers of investment support (Table 7) showed that their fertilization costs per hectare of land follow a pattern similar to that of investment-support users. The greatest increase was recorded in members of the "absolute overinvestment" group (but also in farms at optimum investment levels), and the smallest in the "relative overinvestment" class. Also, just like in other cases, members of the absolute and relative overinvestment groups incurred the greatest costs each year, with the former experiencing much higher growth (54% and 10%, respectively). Nevertheless, an increase in fertilization costs per hectare was recorded in all groups covered by this study in 2010–2019.

Even though the increase in the consumption of plant-protection products is less significant, it follows the same trend as fertilizer use. Therefore, the greatest growth was recorded in members of the "absolute overinvestment" group, and the smallest in the "relative overinvestment" group. Just like in the case of fertilization, the greatest plant-protection costs per hectare are borne by members of the absolute and relative overinvestment groups, which, at the same time (especially in the initial years of the analysis), demonstrate the highest cost intensity of plant protection (Table 6). In turn, the lowest costs per hectare are reported by optimum holdings and go hand in hand with the smallest cost intensity.

## 4. Conclusions

Poland's accession to the EU in 2004 enabled the farms to access a series of support programs under the CAP. The agricultural sector's development capacity mostly relied on those focused on modernization. One of the social and microeconomic goals of support was to make state-of-the-art equipment an enabler of rationalized use of industrial productive inputs. For society, it was important to reduce ground and surface water pollution, adverse impacts on biodiversity, and, last but not least, greenhouse-gas emissions. In turn, the farmers sought the maximization of economic effects (mainly incomes), including by minimizing the cost intensity of production. However, in some cases, the nonrepayable nature of aid contributed to overinvestment, defined as an increase in the assets-to-labor ratio in a situation where the related growth in labor productivity is either absent (absolute overinvestment) or disproportionately small (relative overinvestment). Also, there were farms that did not make any investments, mostly because of their small area and due to the owners' sense of being uncompetitive. Only the situations where the increase in labor productivity was accompanied by growth in the value of assets can be referred to as optimum investment. Therefore, a research project was launched to show the relationship between strategic rationality (reflected by specific investment levels) and operational rationality, represented as the amount of plant fertilization and protection costs per hectare and per PLN 1 worth of production. In all cases covered by this study, i.e., in both users and nonusers of support, the lowest costs intensity of both fertilization and plant protection were recorded by optimum farms, irrespective of whether they did or did not witness an increase in the assets-to-labor ratio. A similar pattern was also found in members of the "relative overinvestment" group, which can be explained by a large scale of investments made within a short time frame. Furthermore, as shown by previous research [57], these farms demonstrate good economic performance. The methodology used in this study to determine the investment levels indicates that they will shift to optimum investment levels in the near future. It is the opposite for members of the "absolute overinvestment" group, although underinvested farms also demonstrated poor performance in that respect. As an equally important aspect from an economic and environmental point of view, high efficiency of plant fertilization and protection usually involved low costs per hectare of agricultural land. A different situation was only found in noninvesting farms. The above means that strategic management (investment) skills go hand in hand with competent operational management (cost optimization). It cannot be fully explained by the fact that state-of-the-art equipment enables a more technologically precise (i.e., more cost effective) use of industrial yield boosters. Indeed, members of the "absolute overinvestment" group also own such equipment but demonstrate the lowest efficiency of all groups covered by the analysis.

Hence, the ultimate reason could be the difference in economic knowledge and farming skills, as mentioned above. This, in turn, provides grounds for certain recommendations for agricultural policy at both the national and EU levels. Even though the patterns analyzed in this paper are found in both users and nonusers of investment support, the grant of that kind of aid should be preceded by a more restrictive analysis of investment efficiency, including in terms of the farm's ability to optimize future operating costs. Moreover, advisory (mostly delivered by public operators) should focus more on the capacity to reduce the costs of using industrial yield boosters with no detriment to yields and production value. The above is an issue of microeconomic importance to producers and of environmental importance to society. The importance of advisory is particularly relevant today when changes are occurring particularly rapidly (many times in the life of one generation), which makes school education insufficient [58]. In the context of the research conducted, attention should be paid to the development of new production technologies (including precision agriculture), which can help in the rational use of yield factors but require large investments. Hence, education and counseling should be comprehensive, covering both agricultural issues (use of yield boosters) and economic issues (appropriate level of investment) in their relationship.

**Author Contributions:** Conceptualization, J.Z., A.S. and N.G.; methodology, J.Z. and A.S.; software, J.Z.; validation, J.Z., A.S. and N.G.; formal analysis, J.Z., A.S. and N.G.; investigation, J.Z., A.S. and N.G.; resources, J.Z.; data curation, J.Z.; writing—original draft preparation, J.Z., A.S. and N.G.; writing—review and editing, J.Z. and N.G.; visualization, J.Z.; supervision, J.Z.; project administration, J.Z.; funding acquisition, J.Z. All authors have read and agreed to the published version of the manuscript.

**Funding:** This research was funded by the National Science Centre, Poland, grant number 2021/41/N/HS4/00443.

**Institutional Review Board Statement:** Not applicable.

**Data Availability Statement:** Not applicable.

**Conflicts of Interest:** The authors declare no conflict of interest.

**Abbreviations**

*ALR*: assets-to-labor ratio; AWU: total labor inputs; EGD: European Green Deal; *EEPP*: environmental efficiency of plant-protection products use; *FC*: fertilization costs; *FE*: fertilization efficiency; *LP*: labor productivity; PLN: polish zloty; *PPC*: plant-protection costs; *SCPP*: consumption scale of plant-protection products.

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
