# Peer review of "Plant Protection and Fertilizer Use Efficiency in Farms in a Context of Overinvestment: A Case Study from Poland"

_agriculture, doi:10.3390/agriculture13081567_

Round 1

Reviewer 1 Report

Comments and Suggestions for Authors

1. The title has no modification and it reflected the content of the manuscript.

2. List of Abbreviations is required 

ALR, AWU, EGD, EEPP, FGDN, LP, PLN, PPC, SCPP

3. The research objective is clear in the Abstract but it is not addressed in the Introduction 1st Paragraph. 

4. The study period was 10 years from 2010 to 2019. The best to calculate the overinvestment status is five separate two-years periods. That's fine. 

4. It is accepted that the production costs have been increased due to the Russian-Ukraine war issues that were indicated by the author's overinvestment because of cost-intensity. 

5. The study did not address the small, medium, and big farmers with their land possession according to which the level of investments were made. The Farm Classification is based on the non-investing farms and investing farms and users of investment and non-users. Figure 1 is the Conceptual Framework of the study or other title Research Procedure Diagram is not clear. The number of farms within the brackets is not clear. Is it the sampling size of the farms? How did you select the number of farms? On what basis? This is not addressed clearly in the manuscript. Take care. Revisit again.

6. All the Tables were calculated by FADN Microdata base. That's fine. Tables 2, 3, 4, 5, 6, and 7 both Fertilization and Plant Protection Years have been indicated for 10 years duration. Kindly check the Table 2 Plant Protection 2019=100?  Check it again in all the Tables.

7. Do you agree that the size of the landholding of the farmers determines the level of the investment in fertilization and plant protection costs?

8. All the citations are numbered in both places. On Page 13 Why did you indicate the Citation is 50 rather than Pawlowski et al. 2021?  

Author Response

Dear Reviewer,

Thank you for your feedback. All of your comments and tips were very helpful and they will certainly help us in preparing future articles as well. We have incorporated the corrections made in the manuscript, and we have included our comments below.

  1. Thank you for appreciating the title.
  2. The list of abbreviations has been added as recommended.
  3. The research objective was reiterated in the penultimate paragraph of the introduction.
  4. We felt that due to possible fluctuations in performance, a 5-year dash for calculating the degree of overinvestment would be effective.
  5. Thank you for your attention regarding figure 1. Once again, the research scheme has been traced. In the name of figure 1 there is information in parentheses "(the number of farms covered by the analysis is specified in brackets).". Renamed "Conceptual framework". The method of selecting the research sample for each type of separate farms was added in the description immediately before figure 1.
  6. Yes, there was an error in Table 2. It should be "2010=100". This has been corrected and the other tables have been checked.
  7. Yes, we agree that size of the landholding of the farmers determines the level of the investment in fertilization and plant protection costs. However, we wanted to investigate whether overinvestment has an impact on the level of fertilization and use of crop protection products, converting this per hectare of farmland. Our other studies show the poor economic situation of farms overinvested absolutely. It turned out that indeed lower fertilizer cost intensity was characteristic of farms investing optimally. From an economic and environmental point of view, high efficiency of plant fertilization and protection usually involved low costs per hectare of agricultural land. A different situation was only found in non-investing farms. The above means that the strategic management (investment) skills go hand in hand with competent operational management (cost optimization).
  8. Citation on page 13 was a mistake. The quote has been corrected and the correct number inserted.

Thank you. We hope that now the manuscript fully meets your expectations.

Kind regards

Reviewer 2 Report

Comments and Suggestions for Authors

The work is interesting but very confusing. Tables should be presented in such a way that it is easier to understand and relate to the text. The introduction presents some phrases that are not supported by the literature and may not be true from a scientific point of view and are concepts of opinion. Example " The climate is....carbon dioxide .... from improper agricultural practice..". This phrase seems to indicate that agriculture and mitrites are almost major causes of global warming. Another phrase " In attempt to solve the above confilts. .... can up with European Green Deal..". In the conflicts mentioned above, he mentions the war in Ukraine that has nothing to do with EGD. The text seems to indicate agriculture as one of the main causes of the greenhouse effect and fertilizers as one of its reasons. Consider this abusive correlation. It is recommended that the text be reformulated and that it may be more related to investments and the use of fertilizer efficiency. The efficiency of fertilizers does not only have to do with overuse, which in terms of agriculture is misuse and cannot be considered as an agricultural practice. Good use will have to do with investments in agricultural education and with the control of phytopharmaceuticals and with investment in techniques for soil mobilization and precision agriculture. I believe that the work should be reformulated and avoid generalist opinions and relate investments in fertilizers with other variables

Author Response

Dear Reviewer,

Thank you for your feedback. All of your comments and tips were very helpful and they will certainly help us in preparing future articles as well. We have incorporated the corrections made in the manuscript, and we have included our comments below.

We agree that the tables and layout of the study may not have been fully clear due to the lack of consistency in the use of nomenclature. The reader could get the impression that the methodology was not followed. Therefore, we decided to supplement the titles of the tables, as well as the description of the tables directly with the names of the indicators from the methodology. In the study conceptual framework, the nomenclature related to environmental efficiency appeared in the final stage. Then, in the description of the method, shortcuts such as EEPP, which did not appear in subsequent sections of the article. This has been corrected in the further part of the work, the nomenclature has been unified and previously used shortcuts have been added to the tables, which should make the article clearer now.

Several items of literature have been added in the introduction. In particular to sentences that were not supported by it. It also clarified issues that allegedly made agriculture the largest emitter of greenhouse gases. Of course, this is not the case, it has been corrected. A note on the importance of agriculture in greenhouse gas emissions is included, as well as an indication of the impact of the use of yield-forming agents on soil quality, relating this to Polish specifics. There was also a separation of two threads, namely a description of the European Green Deal with a brief interjection on the Russian-Ukrainian conflict.

In the final part of the introduction, reference was also made to the investments themselves, overinvestment and the use of fertilizers and plant protection products. In the summary, the importance of education and counseling in improving the efficiency of the use of crop-forming agents and related investments was indicated. Attention was drawn to the need for comprehensive (agricultural and economic) advice, taking into account the interconnectedness of both issues. The challenges of the present day, i.e. the rapid pace of changes, were also pointed out.

Thank you. We hope that now the manuscript fully meets your expectations.

Kind regards

Round 2

Reviewer 2 Report

Comments and Suggestions for Authors

The authors made important efforts to response to thr 

Response to the issues that i put

Comments on the Quality of English Language

The article show important improvement